# A Novel Sol-Gel $Bi_{2-x}Hf_xO_{3+x/2}$ Radiopacifier for Mineral Trioxide Aggregates (MTA) as Dental Filling Materials

Tzu-Sen Yang [1,†], May-Show Chen [2,3,4,†], Cheng-Jyun Huang [5], Chin-Yi Chen [2,5], Agnese Brangule [6], Aleksej Zarkov [7], Aivaras Kareiva [7], Chung-Kwei Lin [2,8,9,*] and Jen-Chang Yang [2,10,11,12,*]

1 Graduate Institute of Biomedical Optomechatronics, Taipei Medical University, Taipei 11031, Taiwan; tsyang@tmu.edu.tw
2 Research Center of Digital Oral Science and Technology, College of Oral Medicine, Taipei Medical University, Taipei 11031, Taiwan; maychen@tmu.edu.tw (M.-S.C.); blachoice@gmail.com (C.-Y.C.)
3 Division of Prosthodontics, Department of Dentistry, Taipei Medical University Hospital, Taipei 11031, Taiwan
4 School of Dentistry, College of Oral Medicine, Taipei Medical University, Taipei 11031, Taiwan
5 Department of Materials Science and Engineering, Feng Chia University, Taichung 40724, Taiwan; chencyi@fcu.edu.tw
6 Department of Pharmaceutical Chemistry, Riga Stradins University, LV-1007 Riga, Latvia; Agnese.Brangule@rsu.lv
7 Institute of Chemistry, Vilnius University, Naugarduko 24, LT-03225 Vilnius, Lithuania; aleksej.zarkov@chf.vu.lt (A.Z.); aivaras.kareiva@chgf.vu.lt (A.K.)
8 School of Dental Technology, College of Oral Medicine, Taipei Medical University, Taipei 11031, Taiwan
9 Additive Manufacturing Center for Mass Customization Production, National Taipei University of Technology, Taipei 10608, Taiwan
10 Graduate Institute of Nanomedicine and Medical Engineering, College of Biomedical Engineering, Taipei Medical University, Taipei 11052, Taiwan
11 International Ph.D. Program in Biomedical Engineering, College of Biomedical Engineering, Taipei Medical University, Taipei 11052, Taiwan
12 Research Center of Biomedical Device, Taipei Medical University, Taipei 11052, Taiwan
* Correspondence: chungkwei@tmu.edu.tw (C.-K.L.); yang820065@tmu.edu.tw (J.-C.Y.)
† The authors contributed equally to the present study.

**Abstract:** Mineral trioxide aggregate (MTA) is well known as an effective root canal filling material for endodontics therapy. Within MTA, bismuth oxide ($Bi_2O_3$) serving as the radiopacifier still has biocompatibility concerns due to its mild cytotoxicity. In the present study, we tried to modify the $Bi_2O_3$ radiopacifier by doping hafnium ions via the sol-gel process and investigated the effects of different doping ratios ($Bi_{2-x}Hf_xO_{3+x/2}$, x = 0–0.3) and calcination temperatures (400–800 °C). We mixed various precursor mixtures of bismuth nitrate ($Bi(NO_3)_3 \cdot 5H_2O$) and hafnium sulfate ($Hf(SO_4)_2$) and controlled the calcination temperatures. The as-prepared Hf-doped $Bi_2O_3$ radiopacifier powders were investigated by thermogravimetric analysis (TGA), X-ray diffraction (XRD), field-emission scanning electron microscopy (FESEM), and transmission electron microscopy (TEM). Portland cement/radiopacifier/calcium sulfate (75/20/5) were mixed and set by deionized water (powder to water ratio = 3:1). Changes in radiopacity, diametral tensile strength (DTS), and in vitro cell viability of the hydrated MTA-like cement were carried out. The experimental results showed that the group containing radiopacifier from sol-gelled Bi/Hf (90/10) exhibited significantly higher radiopacity ($6.36 \pm 0.34$ mmAl), DTS ($2.54 \pm 0.29$ MPa), and cell viability ($84.0 \pm 8.1\%$) ($p < 0.05$) when compared to that of Bi/Hf (100/0) powders. It is suggested that the formation of $\beta$-$Bi_{7.78}Hf_{0.22}O_{12.11}$ phase with hafnium addition and calcining at 700 °C can prepare novel bismuth/hafnium composite powder that can be used as an alternative radiopacifier for root canal filling materials.

**Keywords:** mineral trioxide aggregate; sol-gel; radiopacity; biocompatibility

## 1. Introduction

Root canal treatment is an endodontic treatment sequence for removing the infected pulp and protect the fumigated tooth from future bacterial invasion. Mineral trioxide aggregate (MTA), a powder mixture typically consisting of 75% Portland cement, 20% radiopacifier, and 5% gypsum, has been widely used in endodontic treatments like apexification, direct pulp capping, and root-end filling [1–3]. During application, MTA is mixed with deionized water that can react with Portland cement to produce alkaline calcium hydroxide (CH) and calcium silicate hydrate (CSH) offering hydrated MTA cements unique sealing ability [4], antibacterial properties [5], and preserving pulp vitality [6], whereas the radiopacifier within MTA provide adequate radiopacity to distinguish filling materials from surrounding anatomical structures [7]. Currently, commercially available ProRoot utilizes bismuth oxide ($Bi_2O_3$) as a radiopacifier [8]. Endo CPM Sealer® (EGEO SRL, Buenos Aires, Argentina) [9] and EndoSequence® root repair material (Brasseler USA, Savannah, GA, USA) [10] use barium sulfate ($BaSO_4$) and tantalum oxide ($Ta_2O_5$) respectively to satisfy the radiopacity requirement. Among the reported radiopacifiers for MTA, bismuth oxide ($Bi_2O_3$) exhibited quite a high radiopacity in X-ray photo image contrast for dental clinical uses [11,12].

Although bismuth oxides have been commonly used as radiopacifier within MTA, their intrinsic toxicity, reduced mechanical strength, setting times, difficult manipulation, and tooth discoloration issues have been investigated. Tooth discoloration that might be attributed to the possible reactions of $Bi_2O_3$ with sodium hypochlorite, dentin (collagen), blood contamination (hemoglobin and hematin molecules), or UV-light under anaerobic conditions were extensively addressed [13,14]. Modifying bismuth oxide by adding secondary oxide (i.e., zirconia) via mechanical milling process [15] and wet chemical methods such as precipitation and sol-gel processes [16,17] have been attempted. Zirconia-added bismuth oxide exhibited improved radiopacity performance when compared with the pristine counterpart. Using alternative hydration solutions such as calcium lactate gluconate [18] and silk fibroin solution [19] have been attempted to extend the practical applications of MTA.

Radiopacity usually refers to a material capable of X-ray attenuation relating to the interactions like coherent scattering, photoelectron effect, and Compton scattering between X-ray photons and traversed matter for diagnostic imaging [20]. The photoelectron effect arising from the interaction of X-ray photos with inner-shell electrons is the dominant factor [21]. Radiopacifiers with high atomic number (Z) improve radiopacity by increasing the mass attenuation coefficient ($\mu$) in accordance with the power law of $\mu \propto Z^3$ [22]. It shows almost no scattering radiation and results in a high-quality image.

Hafnium and its alloys showed superior properties in neutrons absorbing, high melting point, and corrosion-resistant [23]. The perspective of applying hafnium oxide ($HfO_2$) in medical science such as neutron detection, bioimplants, biosensors, radiotherapy, and multimodal theranostic imaging [24,25]. An attempt to modify the radiopacity of $Bi_2O_3$ by yttria-stabilized zirconia dopant was reported to show significantly greater radiopacity than that of pure $Bi_2O_3$ [26]. Inspiring by the similar structure in shell electrons but a higher atomic number to zirconium, the study aims to optimize the novel radiopacifier by doping hafnium ions to $Bi_2O_3$ via the sol-gel process for future dental root canal filling materials. The effects of hafnium additions and sol-gel calcination temperatures on the particle morphology, radiopacity, diametral tensile strength, and biocompatibility were investigated to reveal the feasibility of using novel sol-gel $Bi_{2-x}Hf_xO_{3+x/2}$ powder as radiopacifier in MTA.

## 2. Experimental Procedures

### 2.1. Materials

All chemicals used for novel radiopacifier powder fabrication were analytical grade and purchased from Alfa Aesar (Ward Hill, MA, USA) without any further purification. Bismuth(III) oxide ($Bi_2O_3$, 99.9%) (ACROS ORGANICS, New Taipei City, Taiwan), calcium

sulfate ($CaSO_4 \cdot 0.5H_2O$, 97.0%) (Wako Pure Chemical Industries, Osaka, Japan), Portland cement (GoldStar Cement Co., New Delhi, Delhi, India) were used to prepare the MTA-like cement.

### 2.2. Preparation of $Bi_{2-x}Hf_xO_{3+x/2}$ Powders by Sol-Gel Process

In this study, $Bi_{2-x}Hf_xO_{3+x/2}$ powders were produced using the sol-gel process under various thermal conditions. We mixed 10 g of bismuth (III) nitrate pentahydrate ($Bi(NO_3)_3 \cdot 5H_2O$, 98%) with acetic acid ($CH_3COOH$, 99.5%) at 40 °C for 10 min. Then 2-methoxyethanol ($C_3H_8O_2$) was added to form a stable complex without particles sedimentation. Finally, hafnium (IV) sulfate ($Hf(SO_4)_2$, 99.9%) with the x value varied from 0.00, 0.15, 0.20, 0.25, and 0.30 was added to prepare $Bi_{2-x}Hf_xO_{3+x/2}$ in 1 wt% solutions. Here, sample codes for the sol-gelled $Bi_{2-x}Hf_xO_{3+x/2}$ composite powders are listed in Table 1. The powders were collected by an electrostatic dust collector and dried overnight. The harvested products were dried in a vacuum oven at 90 °C overnight and ground into fine powders using a mortar and pestle. The sol-gelled powders after calcining at 400 °C to 800 °C were collected and examined by thermogravimetric analysis, X-ray diffraction (XRD), field-emission scanning electron microscopy (FESEM), and transmission electron microscopy (TEM).

**Table 1.** Sample codes for the sol-gelled $Bi_{2-x}Hf_xO_{3+x/2}$ composite powders calcined at various temperatures for 1 h (CT: calcination temperature).

| x | 0 | 0.15 | 0.20 | 0.25 | 0.30 |
|---|---|------|------|------|------|
| Bi:Hf | 100:0 | 92.5:7.5 | 90.0:10.0 | 87.5:12.5 | 85.0:15.0 |
| CT (°C) | SGB | SGBH075 | SGBH100 | SGBH125 | SGBH150 |
| 400 | SGB_4 | SGBH075_4 | SGBH100_4 | SGBH125_4 | SGBH150_4 |
| 500 | SGB_5 | SGBH075_5 | SGBH100_5 | SGBH125_5 | SGBH150_5 |
| 600 | SGB_6 | SGBH075_6 | SGBH100_6 | SGBH125_6 | SGBH150_6 |
| 700 | SGB_7 | SGBH075_7 | SGBH100_7 | SGBH125_7 | SGBH150_7 |
| 800 | SGB_8 | SGBH075_8 | SGBH100_8 | SGBH125_8 | SGBH150_8 |

### 2.2.1. Thermogravimetric Analysis (TGA)

The differential scanning calorimetry-thermogravimetric analysis (DSC-TGA, Simultaneous SDT 2960, TA instruments Ltd., New Castle, DE, USA) was applied to investigate the thermal stability and phase transformation. The thermal analysis was carried out for heating from 50 °C to 800 °C under 130 $cm^3$/min flowing air environment at a heating rate of 30 °C/min.

### 2.2.2. X-ray Diffraction (XRD)

Powder X-ray diffraction (XRD) analysis was conducted on an SRAM18XHF X-ray powder diffractometer (MacScience Co. Ltd., Kouhoku-ku Yokohama, Japan) with Ni-filtered $CuK_a$ ($\lambda$ =1.542 Å). The operating target voltage and the tube current were 50 kV and 200 mA, respective. Scans were made between 20°~80° at a scanning rate of 2°/min. The diffraction peaks of various specimens were analyzed and corresponding with the Joint Committee of Powder Diffraction Standards (JCPDS) database.

### 2.2.3. FE-SEM and TEM

The morphologies and microstructure were observed using field emission scanning electron microscopy (FE-SEM, JSM-6700F, JEOL, Akishima, Tokyo, Japan) and transmission electron microscopy (TEM, JEOL-2100F, Akishima, Tokyo, Japan). FE-SEM incorporated a cold cathode field emission gun and operated at a voltage range from 0.5–30 kV. High-resolution transmission electron microscopy (HRTEM) and selected area electron diffraction (SAED) studies were carried out at an accelerating voltage of 200 kV.

### 2.2.4. Radiopacity Assay

Various MTA-like cement was produced by blending Portland cement/radiopacitifier agent/calcium sulfate (75/20/5) using a benchtop planetary ball mill (Retsch PM100, Haan, Germany) for 10 min. Each cement was blended at a ratio of 0.3 g powder per 0.1 mL liquid (powder/liquid ratio P/L = 3), loaded into a metal ring (10 mm diameter × 1 mm thickness) fixed with a flat glass plate, and set at 37 °C for 24 h. Six specimen disks (N = 6) were radiographed unless the specimen had flaws. A portable dental X-ray system (VX-65, VATECH, Co., Hwaseong-si, Korea) operating at 62 kV, 10 mA, 0.64 s exposure time, and 30 cm focus-film distance was used. The defect-free specimens were located on occlusal radiographic films (Koadak CR imaging plate size 2; Eastman-Kodak Co, Rochester, NY, USA) and exposed along with an aluminum step-wedge with variable thickness (from 2 to 16 mm in 2-mm increments). The mean gray values of each step of the aluminum step-wedge and the specimens were performed by choosing a region of interest using the iso-density area tool of the imaging processing software, Image J 1.39f (Wayne Rasband, National Institutes of Health, Bethesda, MD, USA).

### 2.2.5. Diametral Tensile Strength (DTS)

After mixing the MTA-like cement with DDW, hydrated cement was loaded into a 6-well cylindrical Teflon mold (6 mm diameter × 5 mm height) and allowed to set at 37 °C with 100% relative humidity in an incubator for 1 day to examine the mechanical properties through DTS testing. The DTS was determined using a universal testing machine (CY-6040A8, Chun Yen testing machines, Taichung, Taiwan) at a crosshead speed of 1.0 mm/min. The DTS value of each cylindrical specimen was calculated from the following equation: $DTS = 2P/\pi bw$ (at the axis of the cylinder); where P is peak load (N), b and w are the diameter (mm) and length (mm) of the specimen, respectively.

### 2.2.6. Cell Viability

The cell viability of the extracted specimens from MTA-like cement with different radiopacitifiers was evaluated with mouse osteoblastic cells (MC3T3-E1) to match the clinical usage for root canal filling materials. The in vitro cell viability was assessed using a methyltetrazolium (MTT) assay according to ISO 10993-Part 5. Prior to cells seeding, a 0.2 g pre-hydrated MTA-like cement disk was mixed with 1 mL fresh medium for 24 h and then the extract medium was filtered through 0.22 μm filters. After incubating the cells with a sterile extract medium at 37 °C for 24 h, cellular responses were quantitatively measured. After the culture medium was removed, 1 mL of the MTT working solution at a concentration of 5 mg/mL in sterile PBS was loaded into each well, then incubated at 37 °C for 3 h. After removing the reagent solution, 1 mL of DMSO was loaded to each well and shaken for 20 min. The absorbance of the resulting solution in each well was acquired immediately at 570 nm using an automated microplate reader (Bio–Tek Instruments, Winooski, VT, USA). All experiments were performed in triplicate (N = 3). The cell compatibility was calculated as follows:

$$\text{Percentage of the control} = \frac{\text{Absorbance of the treated sample}}{\text{Absorbance of the control}} \times 100$$

### 2.2.7. Statistical Analysis

Tukey HSD (honestly significant difference) post-hoc test (online web statistical calculators from the domain name of astatsa.com under https://astatsa.com/OneWay_Anova_with_TukeyHSD/, 26 November 2020) was used to evaluate the statistical significance of the measurement data. The results were claimed statistically different at the level of $p < 0.05$.

## 3. Results and Discussion

### 3.1. Preparation and Characterization of Sol-Gel $Bi_{2-x}Hf_xO_{3+x/2}$ Powders

Among wet-chemical techniques, the sol-gel process is widely used in the preparation of ceramic materials such as metal oxides, carbides, and nitrides due to the advantages over conventional processing technologies in low temperature of the reaction, flexible composition control, high purity, and the ability to develop processes for large-area applications [27]. We prepared $Bi_{2-x}Hf_xO_{3+x/2}$ powders by a sol-gel process.

To explore the processing window for calcination, thermogravimetric analysis (TGA) was performed to reveal the physical and chemical phenomena of the harvested sol-gelled powders upon heating. Figure 1 showed the TGA curve of as-prepared SGBH100 powders as a function of temperature. Upon heating, there was a small weight loss (2.3%) from room temperature to 134 °C that can be attributed to the water evaporation and desorption within the precursors. Drastic weight loss (40.0%) arising from the decomposition of organic compounds occurred from 134 °C up to 357 °C. Thereafter, only trivial weight loss (0.4%) was exhibited to the end of the TGA test. Since the melting point of bismuth oxide is 824 °C, the calcination temperature for the sol-gelled powders is thus set within 400–800 °C at an interval of 100 °C.

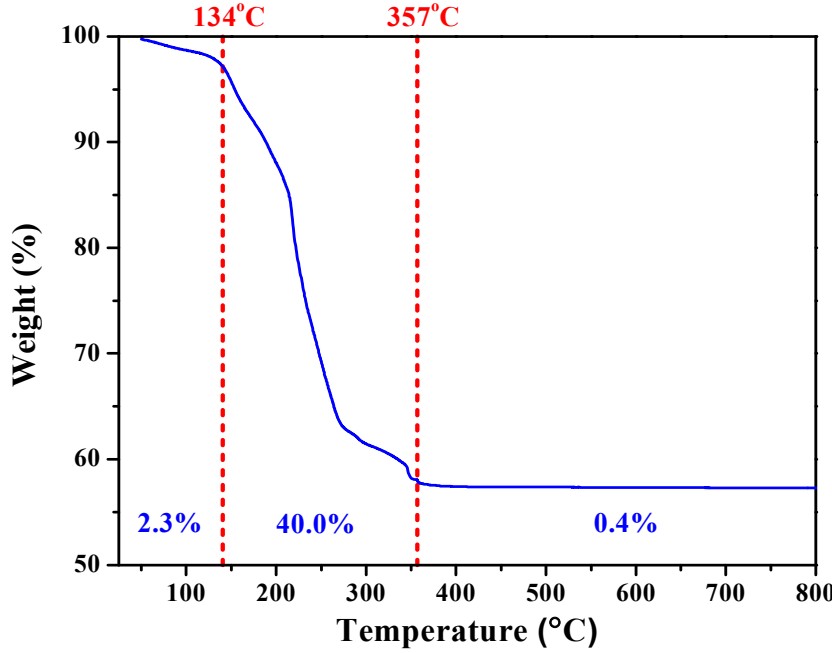

**Figure 1.** Thermal gravimetric (TGA) curve of sol-gelled SGBH100 powders.

Figure 2 showed a typical series of SEM images of SGBH100 powders after calcining at 400–800 °C for 1h. All the calcined powders showed an irregular shape and severe agglomeration of small particles. The particle size of agglomerated powders ranged from 200 nm to 2 μm. The agglomeration seems to increase with increasing calcination temperature. Similar behavior was observed for all the compositions of the sol-gelled powders investigated in the present work. The structure evolution of sol-gelled powders with various hafnium addition and calcining at different temperatures was investigated by the X-ray diffraction (XRD) technique.

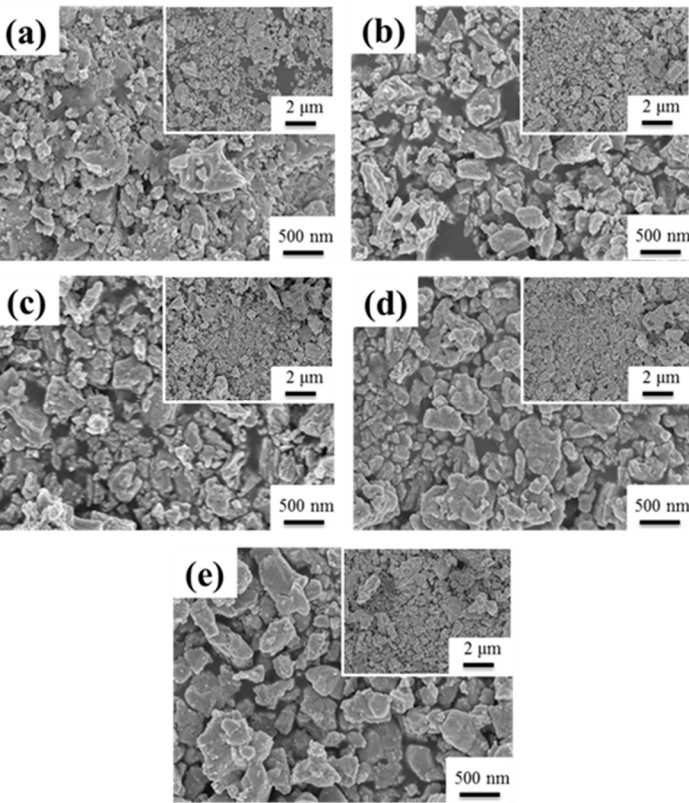

**Figure 2.** The SEM for SGBH100 after calcining at (**a**) 400 °C (**b**) 500 °C (**c**) 600 °C (**d**) 700 °C (**e**) 800 °C for 1 h.

Figure 3 showed the XRD patterns for sol-gelled $Bi_{2-x}Hf_xO_{3+x/2}$ composite powders (x ranged from 0.00 to 0.30, SGB-SGBH150) after calcining at 400–800 °C for 1 h. The as-received sol-gelled powder was amorphous without diffraction peaks. After calcination at a temperature higher than 400 °C, crystalline peaks of various bismuth oxide phases can be observed. As shown in Figure 3a, the pristine sol-gelled powder (SGB series) exhibited a monoclinic $\alpha$-$Bi_2O_3$ phase (JCPDS card No.71-2274). The diffraction peaks' intensities increased with increasing calcination temperature. Bismuth oxides were reported with seven polymorphs, which are monoclinic $\alpha$, tetragonal $\beta$, body-centered cubic $\gamma$, face-centered cubic $\delta$, orthorhombic $\varepsilon$, triclinic $\omega$, and high-pressure hexagonal phase [28]. The $\alpha$-$Bi_2O_3$ is a low-temperature stable phase and can undergo phase transformations to other phases under various thermal conditions and pressures.

With a small amount of hafnium addition (x = 0.15, Bi/Hf = 92.5/7.5, SGBH075), the monoclinic $\alpha$-$Bi_2O_3$ phase was replaced by the formation of metastable $\beta$-$Bi_{7.78}Hf_{0.22}O_{12.11}$ tetragonal phase (JCPDS No. 43-0207) shown in Figure 3b. With an increasing amount of hafnium addition (x = 0.20 to 0.30), in addition to the major $\beta$-$Bi_{7.78}Hf_{0.22}O_{12.11}$ phase, minor $\delta$-$Bi_2O_3$ phase (cubic fluorite structure oxide, JCPDS No. 27-0052) can be observed after calcination at 400–600 °C for 1h. In Figure 3c, the XRD patterns for SGBH100 powders (x = 0.20, Bi/Hf = 90.0/10.0) exhibited a mixture of $\beta$-$Bi_{7.78}Hf_{0.22}O_{12.11}$ and $\delta$-$Bi_2O_3$ phases at a temperature ranged from 400–600 °C. After being treated at 700 °C and above, $\beta$-$Bi_{7.78}Hf_{0.22}O_{12.11}$ became dominant with almost no trace of $\delta$-$Bi_2O_3$ phase. Further increasing the hafnium addition to x = 0.25 (Bi/Hf = 87.5/12.5) showed the similar XRD patterns for SGBH125 powders (Figure 3d) where $\delta$-$Bi_2O_3$ phase was the major phase at low calcination temperature (400–600 °C) and $-Bi_{7.78}Hf_{0.22}O_{12.11}$ became dominant phase at 700–800 °C. The XRD results indicated that sol-gelled pristine bismuth oxide formed a stable monoclinic $\alpha$-$Bi_2O_3$ phase at room temperature. Hafnium additions enable the high temperature tetragonal and cubic phases (i.e., $\beta$-$Bi_{7.78}Hf_{0.22}O_{12.11}$ and $\delta$-$Bi_2O_3$). A high dosage of hafnium addition (x = 0.20 and above) and low calcination temperatures (600 °C

and below) favors the formation of the δ-$Bi_2O_3$ phase. Whereas $Bi_{7.78}Hf_{0.22}O_{12.11}$ exhibited at hafnium addition of x = 0.15 and high calcination temperatures at 700–800 °C.

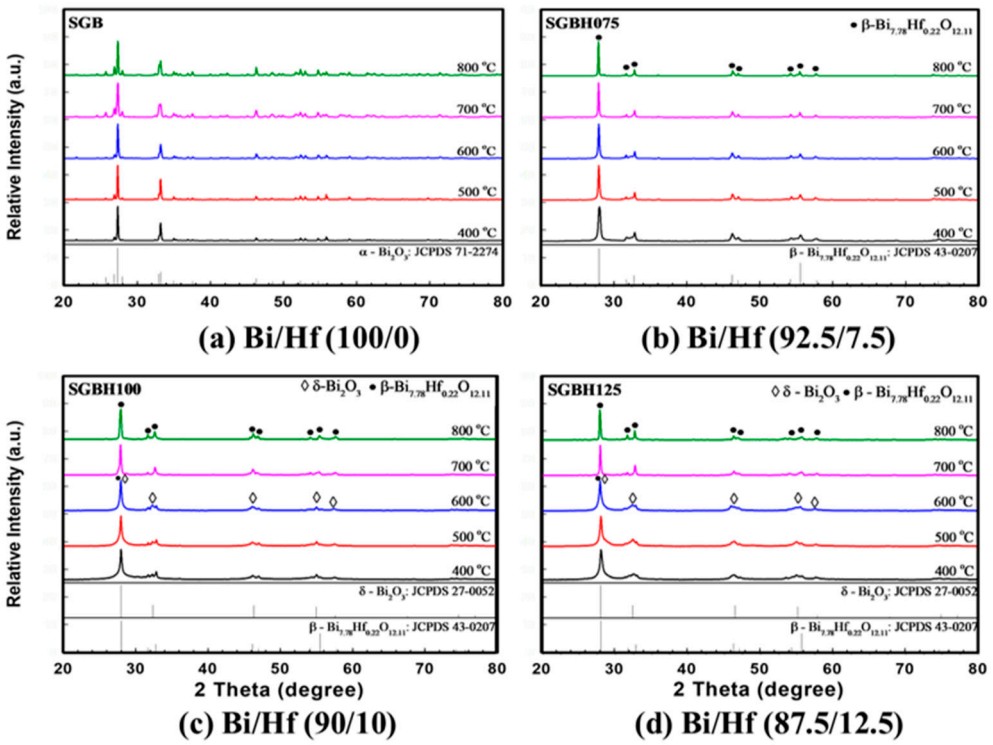

**Figure 3.** The X-ray diffraction (XRD) for various compositions of (**a**) Bi/Hf (100/0) (**b**) Bi/Hf (92.5/7.5) (**c**) Bi/Hf (90/10) (**d**) Bi/Hf (87.5/12.5) after calcination from 400 °C to 800 °C for 1 h.

Helfen et al. reported that the room-temperature monoclinic α-$Bi_2O_3$ transforms to the high-temperature cubic δ-$Bi_2O_3$ upon heating at 729 °C [29]. The mechanochemical process with heating at 820 °C for 24 h then quenching in air showed a stabilized δ-$Bi_2O_3$ structure through the formation of a solid solution with 10 mol % of $HfO_2$, while two β-$Bi_2O_3$ phases were found with 2 mol % of $HfO_2$ dopant [30]. However, our results showed a sol-gel process with a high dosage of hafnium addition (x ≥ 0.20) and low calcination temperatures (≤600 °C) favors the formation of the δ-$Bi_2O_3$ phase. For $HfO_2$–$Bi_2O_3$ prepared by wet precipitation followed by calcination at 600 °C, heating the defect fluorite $Hf_{1-x}Bi_xO_{2-x/2}$ phases (x = 0.40 to 0.75) to temperatures from 700 to 950 °C leads to the formation of pseudocubic $Bi_2Hf_2O_7$, but it decomposes into $HfO_2$ and $Bi_{1.94}Hf_{0.06}O_{3.03}$ with the β-$Bi_2O_3$ structure when heated to 1000 °C [31].

Figure 4 revealed the detailed microstructural examination for SGBH100_7 (x = 0.20, Bi/Hf = 90.0/10.0, calcination at 700 °C) powders by transmission electron microscopy (TEM). As shown in Figure 4a, the irregular shape of powders was noticed. Figure 4b revealed the powders with various sizes (probably due to the agglomeration of a few particles). High-resolution TEM images, Figure 4c, showed the $d_{(201)}$ spacing (3.18 Å) associating with the theoretical value of the β-$Bi_{7.78}Hf_{0.22}O_{12.11}$ phase. Whereas Figure 4d presents the selected area electron diffraction patterns where (201), (002), (220), (222), and (213) diffraction planes corresponding to β-$Bi_{7.78}Hf_{0.22}O_{12.11}$ phase can be identified. The TEM examinations confirmed the XRD results that the β-$Bi_{7.78}Hf_{0.22}O_{12.11}$ is the major crystalline phase in SGHB100 powders after calcining at 700 °C for 1 h.

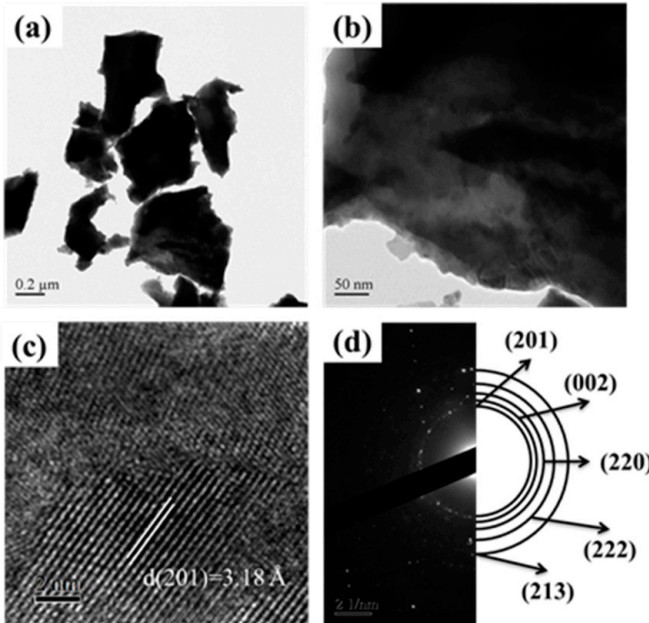

**Figure 4.** The TEM image for Bi/Hf (90/10) sintered at 700 °C (**a**) low magnification TEM (**b**) high magnification TEM (**c**) d-spacing in HRTEM (high resolution transmission electron microscopy) (**d**) SAED (selected area electron diffraction).

### 3.2. Characterization of MTA-Like Cement Comprising $Bi_{2-x}Hf_xO_{3+x/2}$ Radiopacifiers

MTA-like cement primarily consists of inorganic particles of Portland cement and radiopacitifier. After mixing cement powders and water, the hydrated cement creates a calcium silicate hydrate (C-S-H) based impermeable barrier and high radiopacity. Radiopacity is an essential property to offer the relationships of the radiographic interface between dental materials and tooth substrates. Due to the similar radiopacity equivalence between dentin and aluminum (Al) [32], the radiopacity of dental materials was quantitatively assayed using a graduated aluminum step-wedge for measuring its equivalence thickness of optical radiographic densities under X-ray exposure [33]. According to its intended uses, the minimum radiopacity requirement is 3 mmAl for dental root canal sealing materials based on ISO 6876/2001 [34].

Figure 5 presents the radiopacity (mean ± standard deviation (SD) (mmAl)) of hydrated PC associated with various $Bi_{2-x}Hf_xO_{3+x/2}$ radiopacifiers prepared from different Bi:Hf mole ratios and calcination temperatures. It can be noted that the addition of hafnium into sol-gelled bismuth oxide increased the radiopacity performance. The radiopacity of SGB powder was ranged from 4.59 ± 0.19 (SGB_5) to 4.82 ± 0.21 mmAl (SGB_7). Whereas the radiopacity was significantly improved with hafnium addition. Generally, the SGBH100 (i.e., x = 0.20, Bi/Hf = 90.0/10.0) group exhibited the higher radiopacity compared with the other groups. In addition, calcination at 700 °C exhibited the highest radiopacity. All the radiopacity results were summarized in Table 2. Among all the tested sol-gelled powder, the MTA-like cement prepared with SGHB100_7 powder (i.e., Bi/Hf (90/10), calcining at 700 °C) showed the highest radiopacity of 6.26 ± 0.34 mmAl that is significantly different ($p < 0.05$) from that (4.56 ± 0.13 mmAl) prepared by pristine $Bi_2O_3$ powder (i.e., Bi/Hf (100/0)). The experimental error in radiopacity (ranged from 2.4–5.4%) may arise from the sample preparation and X-ray exposure. The increased radiopacity might be attributed to the sol-gel process and the formation of the $\beta$-$Bi_{7.78}Hf_{0.22}O_{12.11}$ dominant phase during calcination. The addition of $Bi_2O_3$ into PC usually resulted in mechanical strength reduction and porosity increase [35]. Using an effective but less amount of radiopacifiers to satisfy the regulatory radiopacity requirement ($\geq$3 mmAl) for dental filling materials might be beneficial to improve their mechanical properties and sealing ability.

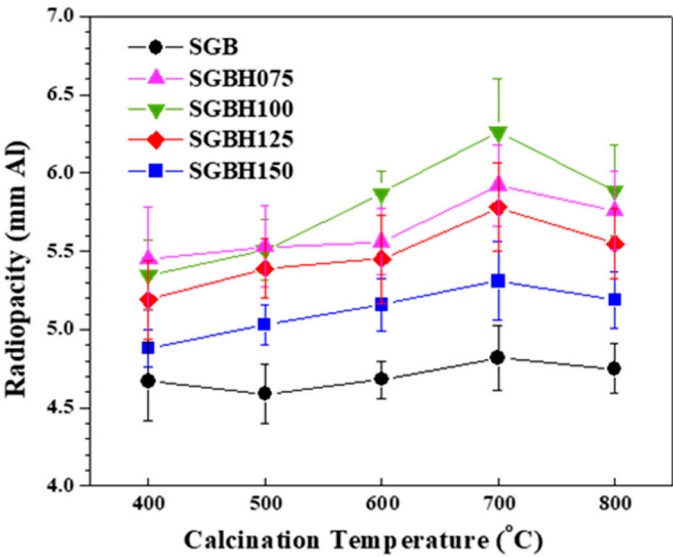

**Figure 5.** The radiopacity for MTA-like cements comprising various radiopacifiers of Bi/Hf (100/0) Bi/Hf (92.5/7.5) Bi/Hf (90/10) Bi/Hf (87.5/12.5) Bi/Hf (85/15) prepared from calcination temperature from 400 °C to 800 °C for 1 h.

**Table 2.** Radiopacities of MTA-like cement prepared by sol-gelled powders. Radiopacities of Portland cement (PC) and PC with commercially available $Bi_2O_3$ are $0.90 \pm 0.15$ and $4.56 \pm 0.13$ mmAl, respectively.

| Calcination Temp (°C) | Means and Standard Deviations of Radiopacity (mmAl) | | | | |
|---|---|---|---|---|---|
| | SGB | SGBH075 | SGBH100 | SGBH125 | SGBH150 |
| 400 | $4.67 \pm 0.25$ | $5.45 \pm 0.33$ | $5.35 \pm 0.22$ | $5.19 \pm 0.25$ | $4.88 \pm 0.12$ |
| 500 | $4.59 \pm 0.19$ | $5.53 \pm 0.26$ | $5.51 \pm 0.19$ | $5.39 \pm 0.19$ | $5.03 \pm 0.13$ |
| 600 | $4.68 \pm 0.12$ | $5.56 \pm 0.21$ | $5.87 \pm 0.14$ | $5.45 \pm 0.28$ | $5.16 \pm 0.17$ |
| 700 | $4.82 \pm 0.21$ | $5.92 \pm 0.26$ | $6.26 \pm 0.34$ | $5.78 \pm 0.28$ | $5.31 \pm 0.25$ |
| 800 | $4.75 \pm 0.16$ | $5.76 \pm 0.25$ | $5.89 \pm 0.29$ | $5.55 \pm 0.22$ | $5.19 \pm 0.18$ |

SGBH100 (Bi/Hf = 90/10, the one with the highest radiopacity and clinical application potential) and its counterpart SGB (Bi/Hf = 100/0) were investigated further by diametral tensile strength and biocompatibility tests. Physical properties usually reflect how materials respond to changes in their usage environments. The mechanical properties might not be crucial because MTA bears no direct occlusal load [36]; however, it is recognized as a representative index for product quality. This test method of the diametral tensile strength (DTS), also known as the splitting tensile strength, is commonly used when conventional tensile testing is difficult to carry out due to the brittle nature of the test specimens. Figure 6 showed DTS values of the MTA-like cement and Portland cement (PC, control group) hydrated using the DDW. After the hydration of 24 h, the MTA-like cement containing sol-gel Bi/Hf (90/10) powders had a DTS value of $2.54 \pm 0.29$ MPa, which was significantly higher than that of Bi/Hf (100/0) powders ($1.78 \pm 0.42$ MPa) but not significantly different from the control group of PC ($p < 0.05$). As to the commercial WMTA (ProRoot MTA, Dentsply/Tulsa Dental, Tulsa, OK) set by DDW, the DTS value of $4.4 \pm 0.1$ MPa was reported by Kao et al. [37]. It is known that particle sizes play a major role in the mechanical properties of cement mortar [38]. All the DTS differences among the experimental groups and commercial MTA might be attributed to the particle fineness and size distribution difference due to agglomeration.

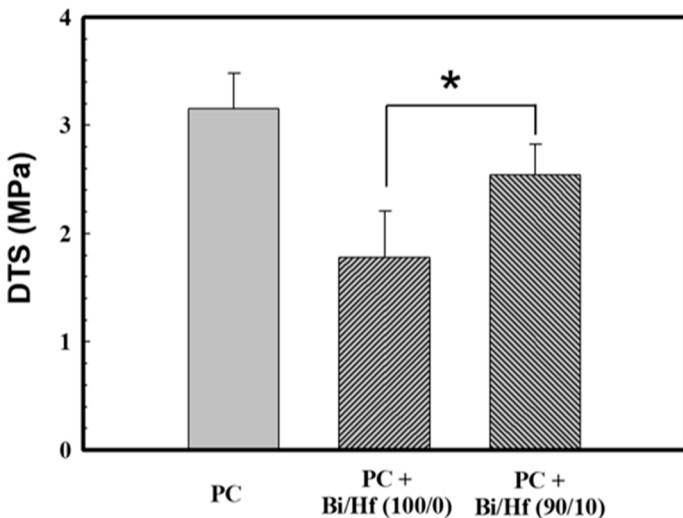

**Figure 6.** The DTS (diametral tensile strength) of MTA-like cement containing radiopacifier of Bi/Hf (100/0) and (90/10), where Portland cement (PC) is the control group. * indicated that these Table 95. confidence interval.

A biocompatibility test is important to investigate the potential adverse tissue responses to biomaterials when applied as intended. Figure 7 presents the cell viability of MC3T3-E1 cells seeded on hydrated MTA-like cement containing sol-gel Bi/Hf (100/0) and (90/10) powders where Portland cement (PC) and polyethylene (PE) are control groups. The cell survival rate for MTA-like cement prepared by Bi/Hf (90/10) powder was $84.0 \pm 8.1\%$ that is statistically higher than that prepared by its counterpart Bi/Hf (100/0) ($75.3 \pm 6.3\%$) ($p < 0.05$), but did not significantly different from both of the control groups ($p > 0.05$).

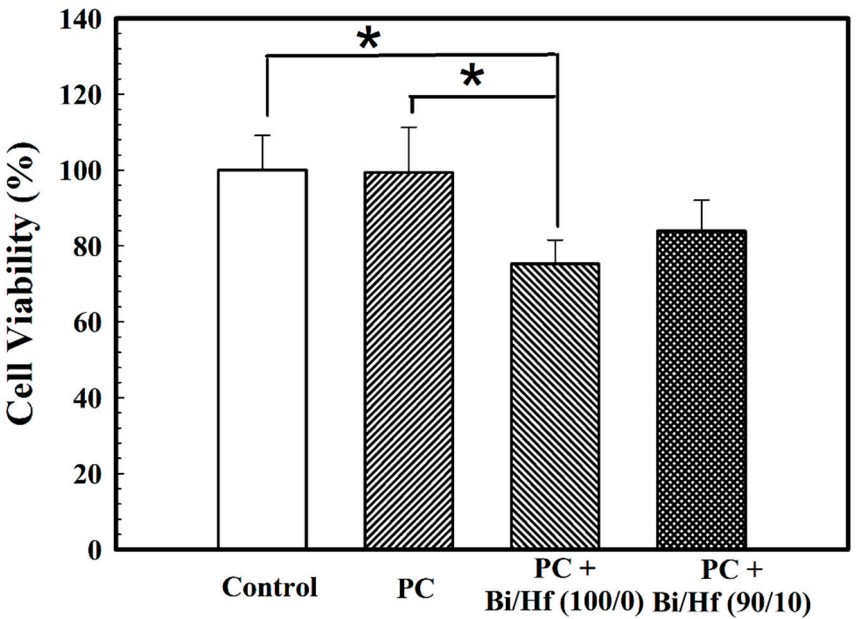

**Figure 7.** Cell viability of MTA-like cement containing different Bi/Hf radiopacifiers where Portland cement (PC) and polyethylene (PE) are control groups. * indicated that these two sets of samples were statistically different at a 95% confidence interval.

As revealed by the above radiopacity examination, diametral tensile strength test, and biocompatibility evaluation, we have demonstrated that sol-gelled $Bi_{2-x}Hf_xO_{3+x/2}$

composite powder with optimized hafnium addition (x = 0.10, Bi/Hf = 90/10, SGBH100) and calcination temperature (700 °C) exhibited superior performance compared to its pristine bismuth oxide counterpart. Investigations concerning tooth discoloration and in vivo animal tests can proceed in future work.

## 4. Conclusions

In this study, the radiopacity for hafnium ion-doped $Bi_2O_3$ powders synthesized using the sol-gel process followed by calcination at various temperatures was performed. The results showed that the SGHB100 (i.e., Bi/Hf (90/10)) powder yields a significant increase of radiopacity over pristine $Bi_2O_3$ (i.e., Bi/Hf (100/0)), indicating new dominant $\beta$-$Bi_{7.78}Hf_{0.22}O_{12.11}$ phase formation as well as possible structure densification by sol-gel process and 700 °C calcination for 1 h. After MTA-like cement hydration, the group containing radiopacifier of sol-gel Bi/Hf (90/10) exhibited the best radiopacity (6.36 $\pm$ 0.34 mmAl), DTS (2.54 $\pm$ 0.29 MPa), and cell viability (84.0 $\pm$ 8.1%) ($p < 0.05$) when compared to that of Bi/Hf (100/0) powders. The Bi/Hf (90/10) powders offer MTA-like cement with high radiopacity, DTS, and acceptable biocompatibility reveal their potential usage as an alternative radiopacifier.

**Author Contributions:** Conceptualization, C.-K.L. and J.-C.Y.; methodology, T.-S.Y. and M.-S.C.; software; validation, A.K.; investigation, C.-J.H., A.B., A.K. and A.Z.; writing—original draft preparation, T.-S.Y. and J.-C.Y.; writing—review and editing, C.-Y.C. and C.-K.L. All authors have read and agreed to the published version of the manuscript.

**Funding:** The authors would like to thank Taipei Medical University Hospital for financially supporting this work under grant no. 110TMU-TMUH-16 and partially supported by MOST 109-2221-E-038-014.

**Institutional Review Board Statement:** Not applicable.

**Informed Consent Statement:** Not applicable.

**Data Availability Statement:** Data is contained within the article.

**Conflicts of Interest:** The authors declare no conflict of interest.

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
