# Peer review of "A Novel Sol-Gel Bi2-xHfxO3+x/2 Radiopacifier for Mineral Trioxide Aggregates (MTA) as Dental Filling Materials"

_applsci, doi:10.3390/app11167292_

Round 1

Reviewer 1 Report

Your paper not improved enough.

Author Response

Not able to response

Reviewer 2 Report

There are some weaknesses through the manuscript which need improvement. Therefore, the submitted manuscript cannot be accepted for publication in this form, but it has a chance of acceptance after a major revision. My comments and suggestions are as follows:

1- Abstract gives information on the main feature of the performed study, but some details about the experimental practice must be added.

2- Authors must clarify necessity of the performed research. Objectives of the study must be clearly mentioned in introduction.

3- The literature study must be enriched. In this respect, authors must read and refer to the following papers: (a) https://doi.org/10.1016/j.jmbbm.2019.02.009 (b) https://doi.org/10.1016/B978-0-12-803581-8.11281-0 However, introduction is too short in the current version.

4- It would be nice, if authors could add some figures (real or schematic) to show concept and some conditions. For instance in subsection 2.2.1 and 2.2.2, and 2.2.4.

5- It is necessary to add the main formula for the discussed aspects. For example, diametral tensile strength. The main reference of each formula must be cited. Moreover, each parameters in equations must be introduced.

6- It would be nice, if authors could presented Fig. 4 in a high quality.

7- Standard deviation is the presented figures must be discussed. In addition, error in calculation must be considered and discussed.

8- In its language layer, the manuscript should be considered for English language editing. There are sentences which have to be rewritten.

9- The conclusion must be more than just a summary of the manuscript. List of references must be updated based on the proposed papers. Please provide all changes by red color in the revised version.

Author Response

Not able to response

Reviewer 3 Report

This research is under the scope of this journal; the topic is relevant for readers, and this research deals with potentially significant knowledge of the field.

However, there are some concerns about the present manuscript:

Introduction

  • What is the new on of this article? Which results are comparable with other studies already done? 
  • Please, correct some typos in the all manuscript.
  • The use “in endodontic treatments” with very good result biological responses to the material. Need references for the sentence to regenerative Endodontic procedures, please read this article was done apical barrier formation and orthograde filling with Calcium silicate cement (CSC materials).https://doi.org/10.3390/jfb10010014 and https://doi.org/10.1016/j.joen.2017.03.005.
  • Discolouration of the tooth
    • Since this type of calcium silicate cement is associated with color change in the medium / long term (https://doi.org/10.3390/app10175793. One of the big problems of the contrast agent (oxide bismuth) on CSC is teeth discolouration with time! (DOI:https://doi.org/10.3390/app10175793; 10.1111/j.1365-2591.2012.02053.x; https://doi.org/10.3390/jfb10010014; https://doi.org/10.1371/journal.pone.0240634
  • The restoration timing on clinical procedures.
    • Other drawbacks of MTA were a lowers values shear bond. One of the major problems with MTA hydraulic cement, in addition to its setting time, is the weak connection to restorative materials, with very low values and unpredictable connections to restoration material (Palma, Materials MDPI, 2018). Did you perform the immediate/delayed adhesive restoration after the placement of the calcium silicate cement? It would be good if you also discussed this point in your discussion. For this, I suggest reading “Effect of restorative timing on shear bond strength of composite resin/calcium silicate-based cements adhesive interfaces.rege; Clin Oral Invest (2020). https://doi.org/10.1007/s00784-020-03640-7. This technique can make it possible to place a permanent restoration in the same session!? Can the authors discuss these points!

(Can discussed the last two points in the future perspectives).

Results

  • Improve the resolution quality of some  figures.
  • “p” or “P” is not standardized. Sometimes is written in capital letters and lower case.

Figures

  • The font / language  in the figure/caption is different from the text. Please, standardized the sized and the font in the figures and charts with the font of the manuscript. 

Discussion 

  • Please, clarified other limitations of this study? 
  • And, clarified the future perspectives.
  •  

    Perhaps in the future, the authors must analyze the colour changes and made shear-bond test on these materials. Because the discolouration is more marked in a medium and long time of evaluation. https://doi.org/10.1371/journal.pone.0240634; https://doi.org/10.1007/s00784-012-0794-1

References

  • The titles of references have a different format, the title of the article is written in capital letters at the beginning of words, others only in lower case.

Author Response

Not able to response

This manuscript is a resubmission of an earlier submission. The following is a list of the peer review reports and author responses from that submission.

Round 1

Reviewer 1 Report

Thank you for submitting the manuscript for review. Please see the detailed critique below:

Highlights:

  1. The topic is relevant for dental clinical application.
  2. The authors have presented valuable data to support majority of their claims.
  3. The methodology is described in detail for replicability.

Caveats

  1. The authors claim that the reason for the study was to develop material that would not stain teeth as it does with Bi2O3, is unsupported.
  2. The author did not show any experiments or data to demonstrate that the dental staining was reduced for Bi/Hf 90/10 as compared to 100/0
  3. There are numerous grammatical and syntax errors which makes it difficult to review the manuscript.

Specific comments:

  1. The authors need to expand statistical analysis section and describe the primary analysis (for example: one-way ANOVA) before Tukey's post hoc analysis. Software used and version along with sample size for each experiment should be provided What did the authors do for repeatability and reproducibility? Who ran which experiments and was the calibration done?
  2. The authors claim that the cell viability was statistical different for Bi/Hf 100/0 vs 90/10 is not real as the standard deviations overlap. Additionally, authors themselves have shown overlapping error for those two groups which does not prove that cell viability of 90/10 is better than that of 100/0. 
  3. The authors need to run additional experiments to support their claims about staining. The two study author cited has used methodologies to test this staining effect. Alternatively, authors can remove this claim for color stain and redraft the manuscript to align with their existing results.

Camilleri J. Staining Potential of Neo MTA Plus, MTA Plus, and Biodentine Used for Pulpotomy Procedures. J Endod. 2015 Jul;41(7):1139-45. doi: 10.1016/j.joen.2015.02.032. Epub 2015 Apr 15. PMID: 25887807.

Felman, Daniel & Parashos, Peter. (2013). Coronal Tooth Discoloration and White Mineral Trioxide Aggregate. Journal of endodontics. 39. 484-7. 10.1016/j.joen.2012.11.053. 

I wish he authors all the luck for redrafting and future projects!

Reviewer 2 Report

  1. I found in your cell viability experiment, you choose mouse osteoblastic cells to be your vitro experiment model. But in ISO10993-part5 Annex C, the protocol chooses the L929. So, why you select MC3T3-E1, and will you continue your research on osteogenesis? Please add your reason in your article.
  2. Please find some references about the solubility of your modified MTA materials and add a solubility test.
  3. In lines 196 and 201, you used twice though not shown here, please give your reference or additional data.
  4. In Figure 6, you are shown the significant difference between group PC+Bi/Hf(100/0) and group (90/10), but what about the group PC and PC+ Bi/Hf(100/0), I also observed that they have difference.
  5. In figure 7, the volume picture of the control group exists error bar. But depending on your calculated formula in 2.2.6, the value of the control group must be 100%.

Round 2

Reviewer 1 Report

Thank you to the authors for their submission and efforts in collecting the data. Although few points were addressed in their response, some items need more clarification or data to support authors' claim. The absence of the discoloration study seriously undermines the rationale of the study. In the present form, there are inconsistencies which add to the confusion and limits the ability to read/understand the manuscript. 

Highlights:

  1. The study deals with an important clinical issue: discoloration of MTA
  2. The study looks at various aspects of material development and diligent data collection

Caveats:

  1. the study does not show discoloration data which is the most important piece of the puzzle
  2. The inconsistencies of terms makes it difficult to follow through the draft. 

Please see my additional comments below and in the attached PDF.

1. TGA results- I want to know how the current standard looks. The sample in isolation seems fine ( no much degradation in the clinically relevant ranges BUT it is hard to conclude that from TGA like this- I would like to see how a current FDA-approved control sample looks, PC?. Also, why chose this particular sample for TGA? How did the others look

2. What is the SD difference between PC and the Bi/Hf samples?

3. The criteria by which you picked and chose only certain samples for certain tests is not at all clear.

4. Cytotoxicity methods are concerning- and not written in detailed.

5. The conclusion is not supported by the data or hypothesis or rationale

Reviewer 2 Report

Your paper not improved enough. And research purpose is not clear.
